# Synthesis and Hydrophilicity Analysis of bis(propane-1,2-diol) Terminated Polydimethylsiloxanes (PDMSs)

**DOI:** 10.3390/ma15030753

**Published:** 2022-01-19

**Authors:** Lan-Hee Yang, Kyeong Eun Park, Sungho Yoon

**Affiliations:** Department of Chemistry, Chung-Ang University, 84 Heukseok-ro, Dongjak-gu, Seoul 06974, Korea; yangd5d5@cau.ac.kr (L.-H.Y.); pke8597@cau.ac.kr (K.E.P.)

**Keywords:** polydimethylsiloxane (PDMS), hydrophilicity, molecular weight, viscosity, contact angle

## Abstract

Among silicone oligomers, polydimethylsiloxane (PDMS) is widely used industrially and has the advantage of improving the properties of other compounds, such as flame-retardant polyurethane (PU). However, as there are barriers to the synthesis of PU-grafted siloxane, owing to the polarity difference between isocyanate and PDMS, numerous research efforts are being aimed at improving the hydrophilicity of PDMS. To improve the hydrophilicity and reactivity of hydroxyl PDMS, bis(propane-1,2-diol)-terminated PDMS (G-PDMS-G) with four hydroxy (-OH) groups was synthesized through ring-opening addition to replace both ends of linear α,ω-hydroxyl PDMS (HO-PDMS-OH) with glycidol, resulting in hydrophilic PDMS rather than dihydroxy PDMS. In all cases of G-PDMS-G, the contact angle and viscosity both decreased by more than 20%, confirming the improved hydrophilicity. In particular, G-PDMS-G-3, which has the largest molecular weight, demonstrated the greatest decrease in viscosity and contact angle (33%).

## 1. Introduction

Polydimethylsiloxane (PDMS) is the most common and widely used silicone compound and is considered an important material, with diverse advantages such as high thermal stability, low glass transition temperature, low surface energy, high molecular flexibility, and biocompatibility. This polymer has been widely used in industry as a lubricant, antifoaming agent, and resin for soft lithography. In addition, it can be applied to improve the properties of other compounds [1,2,3].

One application of PDMS is in flame-retardant polyurethane (PU). This silicone-incorporated PU is non-corrosive, and it is less toxic and generates less smoke when exposed to fire than phosphorus and halogen-based flame retardants [4,5,6]. In addition, it can also be used as a coating agent due to its low surface energy [7,8]. However, PU is a block copolymer with urethane repeats, synthesized by the reaction between isocyanate and polyol molecules, and should contain two or more isocyanate groups and hydroxyl groups for high reactivity. The polar urethane hard segment and non-polar siloxane soft segment in the PU drive phase separation.

Diverse studies on the synthesis of silicone polymers with improved hydrophilicity have been conducted over the last two decades for various purposes [9,10,11,12]. In general, hydrophilic polysiloxanes for perfumes, shampoos, and paint additives are synthesized by a hydrosilylation reaction according to the graft method with a linear copolymer or polyether chain on the PDMS backbone [13,14,15]. In addition, there are reports on the development of biopolymers with hydrophilic groups that are easily biodegradable and non-toxic to humans [16,17]. Other studies demonstrate that increasing the number of hydrophilic saccharides bound to the PDMS backbone or combining oligosaccharides rather than monosaccharides increases hydrophilicity [18,19]. However, when using the graft method, it is difficult to control the branches located along the PDMS backbone. Therefore, Zhang et al. synthesized PDMS with a diol group at one chain end by hydrosilylation of 1,2-allyl glycidol ether using a termination method [20,21].

In this study, bis(propane-1,2-diol)-terminated PDMS (G-PDMS-G), which has a symmetrical structure and includes diol groups at both ends (Figure 1B), was prepared by the terminal method to achieve a higher hydrophilicity and reactivity than those for HO-PDMS-OH (Figure 1A). Additionally, PDMS containing four OH groups was synthesized by a facile post-treatment process involving hydrosilylation, without catalyst, and analyzed for improved hydrophilicity.

## 2. Experimental

### 2.1. Materials

PDMS hydroxyl-terminated (number average molecular weight (Mn): ~550, viscosity: ~25 cSt), PDMS hydroxyl-terminated (viscosity: ~65 cSt), sodium hydride (dry, 90%), glycidol (96%), and tetrahydrofuran (anhydrous, >99.9%, inhibitor-free) were purchased from Merck Korea (Seoul, Korea). PDMS hydroxyl-terminated (weight-average molecular weight (Mw): ~4200) was obtained from Alfa Aesar (Ward Hill, MA, USA).

### 2.2. Sample Characterization

^1^H NMR spectra of the products were measured with a Varian 600 MHz NMR spectrometer (Palo Alto, CA, USA) using CDCl_3_ as the solvent. The molecular weight and polydispersity index (PDI) were determined by gel permeation chromatography (GPC) using a Waters 2410 refractive index detector (Milford, MA, USA) with a Waters 515 HPLC pump. The column was eluted with THF at a flow rate of 1.00 mL/min. GPC curves were calibrated using a polystyrene standard with molecular weights ranging from 162 to 6,570,000 g mol^−1^. Dihydroxy-terminated PDMS sample was coated on a 1.5 cm × 1.5 cm glass slide washed with methanol. The contact angle between the 4 µL DI water droplets and the surface of the modified samples was measured using an SEO Phoenix-300 Touch (Kromtek, Malaysia). The viscosity of the neat samples was measured directly by a CPA-41Z spindle using a Brookfield viscometer (DV2TLVCJ0) (Boston, MA, USA).

### 2.3. Glycidol-Terminated PDMS (G-PDMS-G-1) from HO-PDMS-OH-1

A PDMS hydroxyl-terminated solution (10 g, 18.18 mmol, Mn: ~550, viscosity: ~25 cSt) in THF (100 mL) was cooled in an ice bath under an N2 atmosphere. Then, sodium hydride (36.36 mmol) was slowly added to the solution. The solution, which turned beige with H2 generation, was stirred until clear. Glycidol (28.17 mmol) was injected and reacted for 3 h at room temperature. Distilled water (DW) was used to quench the reaction. The product solution was washed three times with excess DW and the water was removed (yield: 60%).

### 2.4. Glycidol-Terminated PDMS (G-PDMS-G-2) from HO-PDMS-OH-2

PDMS hydroxyl-terminated solution (10 g, 3.030 mmol, viscosity: ~65 cSt) in THF (100 mL) was cooled in an ice bath under an N2 atmosphere. Sodium hydride (6.060 mmol) was injected into the solution and stirred until the solution color changed from beige to clear. Glycidol (12.12 mmol) was added and reacted at room temperature. After 3 h, DW was injected to quench the reaction. Finally, the product solution was washed three times with excess DW and separated (yield: 63%).

### 2.5. Glycidol-Terminated PDMS (G-PDMS-G-3) from HO-PDMS-OH-3

PDMS hydroxyl-terminated solution (10 g, 2.381 mmol, Mw: ~4200) in THF (100 mL) was cooled in an ice bath under an N_2_ atmosphere. Sodium hydride (4.762 mmol) was added to the solution, which turned beige with H_2_ generation, and the solution was stirred until clear. Glycidol (9.524 mmol) was added and reacted for 3 h at room temperature. DW was added to quench the reaction. The product solution was washed three times with excess DW and separated (yield: 65%).

## 3. Results and Discussion

Three types of linear α,ω-dihydroxy PDMS, including HO-PDMS-OH-1 (Mn: ~550, viscosity: ~25 cSt, Sigma Aldrich, St. Louis, MO, USA), HO-PDMS-OH-2 (viscosity: ~65 cSt, Sigma Aldrich), and HO-PDMS-OH-3 (Mw: ~4200, Alfa Aesar), were purchased and used as starting materials. As shown in Figure 2, after the dehydration of both ends of the hydroxy PDMS by NaH, the PDMS anion attacks the secondary carbon of glycidol, and ring opening occurs, producing G-PDMS-G-1, G-PDMS-G-2, and G-PDMS-G-3 [22,23,24].

Figure 3 shows the ^1^H NMR spectra for the structural comparison analysis of the synthesized PDMS. There is no peak in the ^1^H NMR spectrum of HO-PDMS-OH in the 2.5–4.5 ppm range (Figure 3A), unlike in the ^1^H NMR spectrum of glycidol (Figure 3B). However, five chemical shifts were observed in the range of 2.5–4.5 ppm of the ^1^H NMR spectra of G-PDMS-G-1, G-PDMS-G-2, and G-PDMS-G-3, in which both ends were substituted with glycidol (Figure 3C–E). As shown in Figure 3C, two pairs of geminal protons (H_f_/H_g_, H_i_/H_j_) on the secondary carbon out of the five protons, excluding the hydroxyl group, were produced as diastereotopic protons because of the tertiary carbon substituted on the adjacent carbon, which is the chiral center [25,26]. As in the case of glycidol, both pairs of geminal protons (H_a_/H_b_, H_d_/H_e_) are diastereotopic protons (Figure 3B). The chemical shift observed in the 2.5–4.5 ppm range, was not the same as that representing the end of the substituted PDMS. It is likely that glycidol is asymmetrical, with two terminal OH groups, such as the final G-PDMS-G structure shown in Figure 2. However, it could also be symmetrical and a structure having both. While it is difficult to predict the exact structure of the modified glycidol, the C NMR results in Appendix A, from Supplementary Material, point toward the asymmetrical version with the terminal OH groups.

The experimental ^1^H NMR and GPC results for the dihydroxy-terminated PDMS (HO-PDMS-OH-1, 2, and 3) starting materials and the PDMS glycidol substitution products with four OH groups (G-PDMS-G-1, 2, and 3) are shown in Table 1. The Mn of the starting PDMS was calculated from the ^1^H NMR results for the end group. The Mn, Mw, and PDI of the starting and product PDMS molecules were determined using GPC. Considering only the terminal change, the molecular weight was expected to increase by 148.16 after the two glycidols were added via substitution. However, the product demonstrated a greater increase in molecular weight. A side reaction produced by the siloxane anion generated by the terminal substitution process of hydroxyl-terminated PDMS appears to have caused disproportionation, resulting in a widening of the molecular weight distribution [27,28].

The water contact angle was measured to confirm the hydrophilicity generated by substituting glycidol at the end of linear HO-PDMS-OH. The HO-PDMS-OH and G-PDMS-G samples were spin-coated on slide glass at 5000 rpm for 40 s without curing, which could have caused molecular changes. Figure 4A,B are photographs of a water droplet on the surfaces of HO-PDMS-OH-1 and G-PDMS-G-1, which show that the contact angle has changed significantly. Figure 4C is a graph showing the droplet contact angle of each siloxane polyol. These data indicate that the contact angle increases as the molecular weight increases in the case of HO-PDMS-OH (48.1 ± 0.45, 58.9 ± 6.8, and 64 ± 0.3). The contact angles of G-PDMS-G-1, 2, and 3 were 36.6 ± 0.9, 41.4 ± 1.3, and 42.7 ± 4.25. These angles are smaller than those of HO-PDMS-OH-1, 2, and 3 by 11.5, 17.5, and 21.3, respectively. Therefore, the hydrophilicity improved with increasing OH concentration.

The viscosity changes of HO-PDMS-OH and G-PDMS-G upon glycidol substitution are presented in Table 2. The contact angle data for hydroxy-terminated PDMS indicate that longer HO-PDMS-OH siloxane chains correlate with higher hydrophobicities and viscosities. All G-PDMS-G viscosities decreased by 22.77, 34.0, and 88.36 cP compared to those of HO-PDMS-OH, and the sample with the highest molecular weight exhibited a high viscosity difference. Therefore, glycidol substitution is assumed to increase the hydrophilicity. Although the diol substituted hydroxy terminal structure and the increased hydrophilicity were confirmed, the expected molecular weight reproduction and accurate end group analysis were insufficient, and further research is needed.

## 4. Conclusions

In this study, G-PDMS-G-1, G-PDMS-G-2, and G-PDMS-G-3 were synthesized with four OH groups by diol substitution at both ends of hydroxy-terminated PDMSs (HO-PDMS-OH-1, HO-PDMS-OH-2, and HO-PDMS-OH-3). The ^1^H NMR results confirmed that glycidol was substituted at both ends. In addition, the water contact angles of G-PDMS-G on both sides decreased by 24%, 30%, and 33%, respectively, compared to those of dihydroxyl PDMS. In addition, the viscosities of G-PDMS-G also decreased compared to those of HO-PDMS-OH in all cases, by 24%, 30%, and 33%. Based on these results, it was concluded that increased OH concentration improves hydrophilicity. Therefore, G-PDMS-G with high hydrophilicity is expected to be useful in the field of PU synthesis, in which flame retardancy is achieved by reacting with diisocyanate.

## Figures and Tables

**Figure 1 materials-15-00753-f001:**
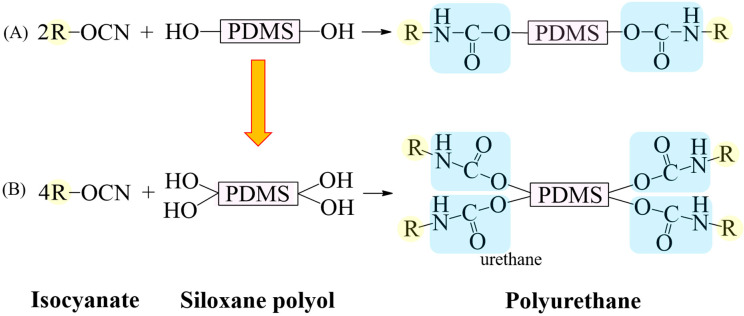
Schematic diagram of reaction between isocyanate and siloxane polyol for synthesis of PU. (**A**) Synthesis of PU using siloxane polyol with two hydroxyl groups, producing two urethane groups. (**B**) Synthetic reaction from a siloxane polyol with four hydroxyl groups, resulting in four urethane groups.

**Figure 2 materials-15-00753-f002:**
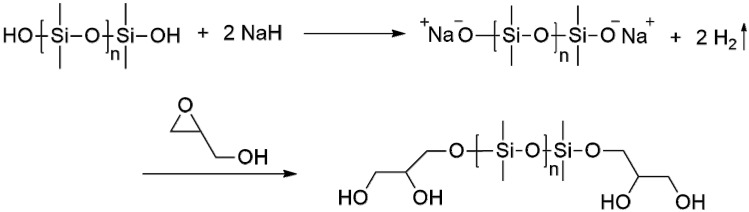
Reaction of dihydroxyl-terminated PDMS (HO-PDMS-OH) to glycidol-terminated PDMS (G-PDMS-G) with sodium hydride.

**Figure 3 materials-15-00753-f003:**
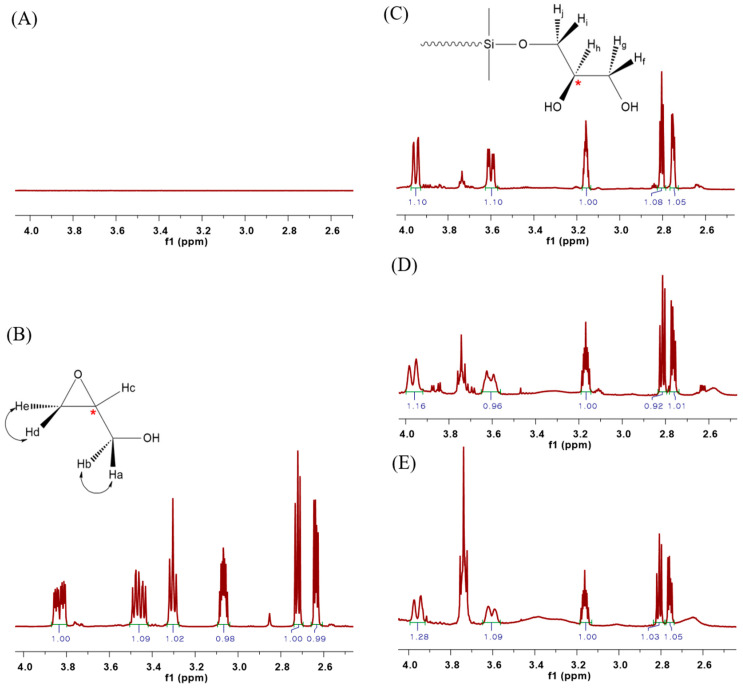
^1^H NMR spectra of (**A**) HO-PDMS-OH, (**B**) glycidol, (**C**) G-PDMS-G-1, (**D**) G-PDMS-G-2, and (**E**) G-PDMS-G-3.

**Figure 4 materials-15-00753-f004:**
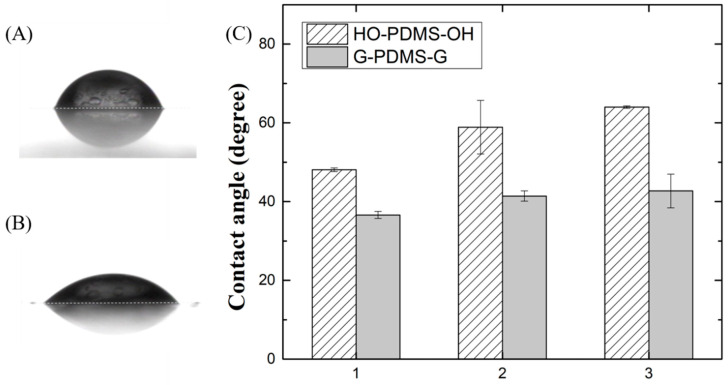
Photo of water droplets on (**A**) HO-PDMS-OH-1 coated surface and on (**B**) G-PDMS-G-1 coated surface. (**C**) Graph showing contact angle of water droplet on HO-PDMS-OH and G-PDMS-G.

**Table 1 materials-15-00753-t001:** Molecular weights of dihydroxy-terminated PDMS (OH-PDMS-OH-1, 2, and 3) and glycidol-terminated PDMS (G-PDMS-G-1, 2, and 3) by ^1^H NMR and GPC.

Entry	Mn (NMR)	Mn (GPC)	Mw (GPC)	PDI (GPC)
HO-PDMS-OH-1	438	686	804	1.17
HO-PDMS-OH-2	3295	3021	4368	1.51
HO-PDMS-OH-3	3450	3043	4902	1.61
G-PDMS-G-1		883	969	1.10
G-PDMS-G-2		6499	9671	1.49
G-PDMS-G-3		6712	8831	1.32

**Table 2 materials-15-00753-t002:** Viscosities of dihydroxy PDMS and glycidol-terminated PDMS.

	HO-PDMS-OH		G-PDMS-G	
	Viscosity (cP)	Temperature (°C)	Viscosity (cP)	Temperature (°C)
1	34.74	21.7	11.97	21.8
2	98.10	21.4	53.1	21.4
3	109.2	21.4	20.84	21.2

## Data Availability

The data presented in this study are available upon request from the corresponding author.

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
