# Peer review of "Synthesis and Hydrophilicity Analysis of bis(propane-1,2-diol) Terminated Polydimethylsiloxanes (PDMSs)"

_materials, 2022, doi:10.3390/ma15030753_

Round 1
Reviewer 1 Report
In this paper, authors reported synthesis of bis(propane-1,2-diol) terminated polydimethylsiloxanes (PDMSs) and evaluated it using NMR, GPC and its contact angle.
Some of my concerns are:
1. It would be nice to have evaluation with an application. Authors mentioned one of the targets is to synthesis flame retardant PU. Authors mentioned their future research plan (e.g.L.163-166) but I am wondering what would be the other challenges to proceed further into application and its evaluation (microscale combustion calorimetry etc.)
2. I am wondering what would be the contact angle after curing. Also are there any error bars in contact angle measurement (Fig. 4).
3. Have you evaluated the sample with other characterization methods too (e.g. FTIR.)
Overall, the report shows the high hydrophilicity of their sample, however, it would be nice to have evaluation with its application . It would be nice to mention the future plan.
Reviewer 2 Report
1. This study reports the synthesis of bis(propane-1,2-diol)-terminated PDMS (G-PDMS-G) from α,ω-hydroxyl PDMS, which pursues enhancing hydrophilicity of PDMS. Hence the title does not precisely reflect this theme.
2. Figure 4(c) requires error bars to demonstrate the uncertainty ranges of measurement.
3. “longer HO-PDMS-OH siloxane chains” in the second sentence below Table 2 is incorrect since HO-PDMS-OH has a smaller size than G-PDMS-G obviously. The decrease in viscosity after increasing hydrophilicity is deemed as the response to the change in molecular packing manner.
4. It is strongly suggested that a comparison of SEM images better in submicron scale of the surfaces of the two samples where the water-contact angles are measured should be included in the manuscript.
5. If possible, measurement of particle size distributions of the very dilute dispersions of HO-PDMS-OH and G-PDMS-G in water, respectively, by the dynamic light scattering approach will largely increase citation of this work.
Reviewer 3 Report
Comment:Manuscript is well written,good sound and so interesting.
Although I had read manuscript several times, there is only just one comment is that Refs should to be updated.
Author Response
We thank the reviewer for pointing out, We recently updated the references.
Reviewer 4 Report
This manuscript by Yang et al. describes the synthesis of bis(propane-1,2-diol) terminated polydimethylsiloxanes, as soft segment components in polyurethanes. The increased hydrophilicity offered by additional hydroxyl groups reduces the phase separation that can be a problem during polyurethane synthesis using (hydrophilic) di-isocyanate and (hydrophobic) oligosiloxane units.
1) The problem to be addressed was clearly stated. Hydrophilic di-isocyanate units can phase separate from hydrophobic oligodimethylsiloxane units, which can impede the polymerisation reaction to prepare silicone-containing polyurethanes with flame retardant properties.
2) The approach taken (to add more hydroxyl-containing units at the ends of the siloxane units) appears to be novel and provides an improvement (increased hydrophilicity, as demonstrated by the decrease in contact angle).
3) Clearly that is relevant to polyurethane producers, which makes it interesting to polyurethane chemists.
4) The manuscript is well written. The data is presented clearly and supports the conclusions.
5) The authors appear to have covered relevant literature adequately. I note that the most recent reference is from 2016, but there appears to have been no pertinent publications since that date. (Polyurethane materials have been around for over half a century and much of the relevant literature is likely to be of a similar vintage.)
As the manuscript is well written and I could find no fault in the scientific content, I am happy to recommend publication in its present form.
Author Response

(The authors gave the same response as above.)

Round 2
Reviewer 1 Report
Authors have addressed all of my concerns with the original manuscript. Thus, I support publication.